# Clathrin modulates vesicle scission, but not invagination shape, in yeast endocytosis

**Wanda Kukulski[1,2,3]\*[†], Andrea Picco[2,4][†], Tanja Specht[2], John AG Briggs[1,2][‡], Marko Kaksonen[1,2,4][‡]**

[1]Structural and Computational Biology Unit, European Molecular Biology Laboratory, Heidelberg, Germany; [2]Cell Biology and Biophysics Unit, European Molecular Biology Laboratory, Heidelberg, Germany; [3]Division of Cell Biology, MRC Laboratory of Molecular Biology, Cambridge, United Kingdom; [4]Department of Biochemistry, University of Geneva, Geneva, Switzerland

**Abstract** In a previous paper (*Picco et al., 2015*), the dynamic architecture of the protein machinery during clathrin-mediated endocytosis was visualized using a new live imaging and particle tracking method. Here, by combining this approach with correlative light and electron microscopy, we address the role of clathrin in this process. During endocytosis, clathrin forms a cage-like coat around the membrane and associated protein components. There is growing evidence that clathrin does not determine the membrane morphology of the invagination but rather modulates the progression of endocytosis. We investigate how the deletion of clathrin heavy chain impairs the dynamics and the morphology of the endocytic membrane in budding yeast. Our results show that clathrin is not required for elongating or shaping the endocytic membrane invagination. Instead, we find that clathrin contributes to the regularity of vesicle scission and thereby to controlling vesicle size.

**\*For correspondence:** kukulski@mrc-lmb.cam.ac.uk

[†]These authors contributed equally to this work
[‡]These authors also contributed equally to this work

**Competing interests:** The authors declare that no competing interests exist.

## Introduction

Clathrin-mediated endocytosis is a conserved cellular process for uptake of nutrients and internalization of cell surface components. The cage-forming protein clathrin is a major coat component thought to scaffold membrane and protein machinery during vesicle formation. In yeast, clathrin contributes to actin-dependent endocytosis and is present at all endocytic sites but is not essential for endocytic uptake (*Newpher et al., 2005*; *Chu et al., 1996*; *Payne et al., 1988*). Previous research has shown that upon deletion of clathrin heavy chain, there is a 73% reduction in the frequency of endocytic vesicle budding events (*Kaksonen et al., 2005*; *Newpher and Lemmon, 2006a*), which can explain the reduced rate of receptor-mediated uptake (*Chu et al., 1996*; *Payne et al., 1988*). The clathrin heavy chain deletion phenotype can be partially rescued by over-expression of clathrin light chain (*Newpher et al., 2006b*). These findings suggest a role for clathrin in efficiently initiating vesicle budding. Although deletion of clathrin heavy chain also causes a strong growth phenotype and accumulation of internal membrane compartments (*Payne et al., 1988*; *Payne and Schekman, 1985*), the reduction of receptor-mediated endocytosis is independent of these effects (*Tan et al., 1993*). The abnormal morphology and slow growth of clathrin-deficient cells is likely a downstream, cumulative effect of disrupted clathrin-mediated vesicle formation at the trans-Golgi network, endosomes and plasma membrane (*Newpher et al., 2005*; *Brodsky et al., 2001*).

Taken together, previous studies suggest that clathrin plays a role in initiating vesicle budding. However, roles for clathrin at stages following initiation are not clear. Clathrin is able to induce membrane curvature in vitro (*Dannhauser and Ungewickell, 2012*), but the distribution of endocytic invagination depths as well as the localization of endocytic coat components observed by immuno-electron microscopy do not change significantly upon clathrin deletion, suggesting that clathrin does not define membrane morphology in vivo (*Idrissi et al., 2012*). We have previously proposed that clathrin assembles on a flat membrane and only forms a curved lattice upon actin-induced membrane bending in yeast (*Kukulski et al., 2012a*). In mammalian cells, it was recently shown that the curvature of the clathrin coat changes during vesicle budding (*Avinoam et al., 2015*). Thus, there is growing evidence that the clathrin coat is not a rigid scaffold that imposes shape to the forming endocytic vesicle.

We have previously used live cell imaging and correlative light and electron microscopy to generate a detailed quantitative description of assembly dynamics of the endocytic protein machinery and the corresponding membrane morphology changes in budding yeast (*Picco et al., 2015*; *Kukulski et al., 2012a*). Here, we have applied these methods to budding yeast cells in which the gene encoding clathrin heavy chain has been deleted (*chc1Δ*) to clarify the role of clathrin during membrane reshaping and vesicle formation.

## Results and discussion

We first asked how the dynamics of membrane invagination are affected by the absence of clathrin. The Sla1 protein assembles into the endocytic coat before the membrane starts bending and then moves in at the tip of the membrane invagination (*Picco et al., 2015*; *Idrissi et al., 2008*). Although fewer endocytic sites are observed in *chc1Δ* cells, Sla1 marks those that successfully initiate budding (*Kaksonen et al., 2005*; *Newpher and Lemmon, 2006a*). The distribution of Sla1 on the invagination is not altered by the deletion of clathrin (*Idrissi et al., 2012*). The movement of the Sla1-GFP centroid, therefore, acts as a marker for the growth of the endocytic membrane invagination (*Picco et al., 2015*). We tracked the fluorescence intensity and movement of Sla1-GFP in living wild-type and *chc1Δ* cells (*Figure 1A* and *Figure 1—figure supplement 1*). The total lifetime of Sla1-GFP at the endocytic sites was shortened in *chc1Δ* cells, due to an accelerated assembly phase, as described previously (*Kaksonen et al., 2005*; *Newpher and Lemmon, 2006a*), and Sla1-GFP disassembly at the end of the endocytic process was subtly slowed down (*Figure 1—figure supplement 1*). The Sla1-GFP centroids moved slightly further into the cell than in wild-type cells (*Figure 1B*), probably as a consequence of the slowed disassembly of Sla1-GFP that allowed us to track the patches in the cytoplasm for longer. The rate of the Sla1-GFP centroid movement in *chc1Δ* cells was indistinguishable from that of Sla1-GFP in wild-type cells (*Figure 1A and B*). Thus, the speed at which the invagination elongates was not affected by the lack of clathrin.

We then asked if scission of the vesicle is affected by the absence of clathrin. Rvs167 is an amphiphysin-like BAR domain protein that assembles at the tubular part of the invagination and regulates vesicle scission (*Picco et al., 2015*; *Kukulski et al., 2012a*; *Youn et al., 2010*; *Smaczynska-de Rooij et al., 2012*; *Kishimoto et al., 2011*). In wild-type cells, the peak in Rvs167-GFP fluorescence intensity coincides with scission (*Kukulski et al., 2012a*) and the rapid directional disassembly of Rvs167 after scission leads to a fast inward movement of the Rvs167-GFP centroid (*Picco et al., 2015*). Rvs167-GFP movement and peak fluorescence intensity therefore act as markers for vesicle scission in wild-type cells. The Rvs167-GFP centroid trajectories in *chc1Δ* cells often did not display the characteristic fast inward movement observed in wild-type cells, but appeared highly irregular (*Figure 1C*). Due to this irregularity, we could not align and average the Rvs167-GFP centroid movements as we did for Sla1-GFP. When we examined the fluorescence intensity of Rvs167-GFP at the endocytic sites, we found that the time taken to assemble Rvs167-GFP molecules at the endocytic site was similar in both *chc1Δ* and wild-type cells (*Figure 1D*). However, after the assembly phase, the fluorescence intensity of Rvs167-GFP persisted significantly longer in *chc1Δ* cells than in wild-type cells (*Figure 1D*). These findings suggest that the disassembly of Rvs167 molecules and thereby the regulation of vesicle scission are impaired in *chc1Δ* cells.

To directly visualize the effects of clathrin absence on membrane reshaping, we applied correlative fluorescence microscopy and electron tomography. We thereby located and imaged endocytic intermediates in *chc1Δ* cells expressing Sla1-GFP as well as Abp1-mCherry, which we used as a

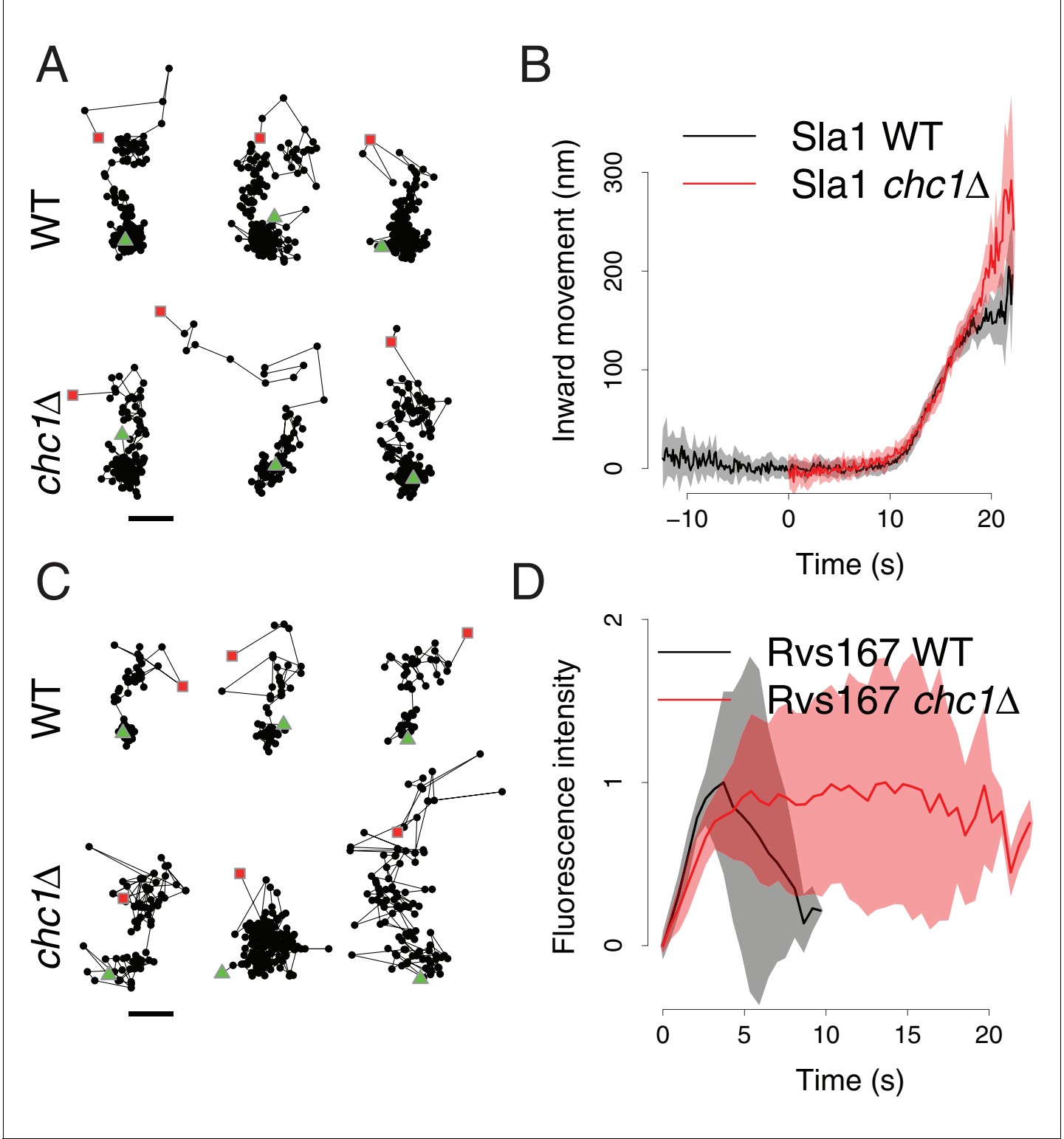

**Figure 1.** Sla1 and Rvs167 dynamics in *chc1Δ* cells. (A) Representative examples of Sla1-GFP trajectories from wild-type (upper row) and *chc1Δ* cells (lower row). Green triangles and red squares mark start and end points of the trajectories, respectively. Scale bar is 100 nm. (B) Average Sla1-GFP inward movement over time, in wild-type cells (black line, n = 50 trajectories) and *chc1Δ* (red line, n = 41 trajectories). Shadings indicate the 95% confidence interval. (C) Representative examples of Rvs167-GFP trajectories from wild-type (*Picco et al., 2015*) (upper row) and *chc1Δ* cells (lower row). Green triangles and red squares mark start and end points of the trajectories, respectively. Scale bar is 100 nm. (D) Average Rvs167-GFP fluorescence intensity over time, in *chc1Δ* (red line, n = 56 trajectories) and wild-type cells (black line, n = 54 trajectories). Fluorescence intensity profiles have been

*Figure 1 continued on next page*

*Figure 1 continued*

normalized between 0, which corresponds to their minimum in intensity, and 1, which corresponds to their maximum in intensity. Shadings indicate the standard deviations from the mean.

The following figure supplement is available for figure 1:

**Figure supplement 1.** Sla1-GFP fluorescence intensity in *chc1Δ* cells.

marker for the presence of invaginations or vesicles (*Kukulski et al., 2012a*). We reconstructed electron tomograms of 59 endocytic sites marked by Abp1-mCherry, or by both Abp1-mCherry and Sla1-GFP (*Figure 2A*, see also 'Materials and methods'). From these data, we extracted membrane profiles of invaginations and analyzed their shapes. We measured invagination depth, curvature of invagination tips (*Figure 2B*), appearance of tubular segments and position of the neck (*Figure 2—figure supplement 1*). All parameters showed a similar distribution as in wild-type cells (*Kukulski et al., 2012a*). We next measured the axes of ellipsoids fitted to endocytic vesicles. We calculated vesicle surface areas and found that on average they were similar to wild-type, but the variance was significantly larger (*chc1Δ*: mean 7'291 nm$^2$, SD 4'373 nm$^2$, n = 51. wt: mean 6'380 nm$^2$, SD 1'929 nm$^2$, n = 62. p<0.0001 for variances) (*Figure 2C*, see also 'Materials and methods'). Thus, while invaginations were unaffected, vesicle sizes were significantly more variable in absence of clathrin.

Taken together, our data shows that the rate at which the invagination (marked by Sla1-GFP) grows, as well as the detailed morphology of the membrane invaginations observed by electron tomography, were the same in wild-type and *chc1Δ* cells. We thus conclude that the invagination process is unaltered, and neither elongation nor shaping of the invagination requires clathrin. Clathrin has been shown to induce membrane curvature in vitro (*Dannhauser and Ungewickell, 2012*), and we cannot rule out that it has a membrane sculpting role in other vesicle budding events, for instance when the membrane is subjected to less pressure than the turgor of the yeast cell. Furthermore, we observed that disassembly of Rvs167-GFP, which is a marker for scission in wild-type cells, became highly irregular in *chc1Δ* cells and the resulting vesicles were significantly more variable in size. Since invagination morphologies were unchanged, it is likely that the here described variability of vesicle sizes is related to a loss of regulation of the position of the scission site or of the correct timing of the scission event. How clathrin modulates disassembly of Rvs167 is an open question: The two proteins occupy adjacent regions on the endocytic invagination (*Idrissi et al., 2008*), and in vertebrates, amphiphysins bind directly to clathrin and the endocytic adaptor AP-2 (*McMahon et al., 1997*; *Ramjaun and McPherson, 1998*; *Slepnev et al., 2000*). Thus, we speculate that protein-protein interactions involving clathrin could modulate the disassembly dynamics of Rvs167, either through direct binding or recruitment of other regulatory proteins.

Previous work showed that clathrin has a role in initiating the vesicle budding process. Our study shows it has an additional role in modulating endocytic protein disassembly and the timing or position of the scission event and thereby the sizes of the resulting endocytic vesicles. We found no role for clathrin in sculpting the membrane during endocytic invagination in budding yeast.

## Materials and methods

### Genotypes of *Saccharomyces cerevisiae* strains used in this study

Yeast strains were generated using the toolbox described by *Janke et al., 2004*. Strains were maintained as heterozygous diploids to minimize generation of suppressors of the clathrin deletion mutation, as described in (*Kaksonen et al., 2005*).

#### For correlative microscopy

*chc1Δ*, Sla1-EGFP, Abp1-mCherry (*Kaksonen et al., 2005*) (MKY2800):

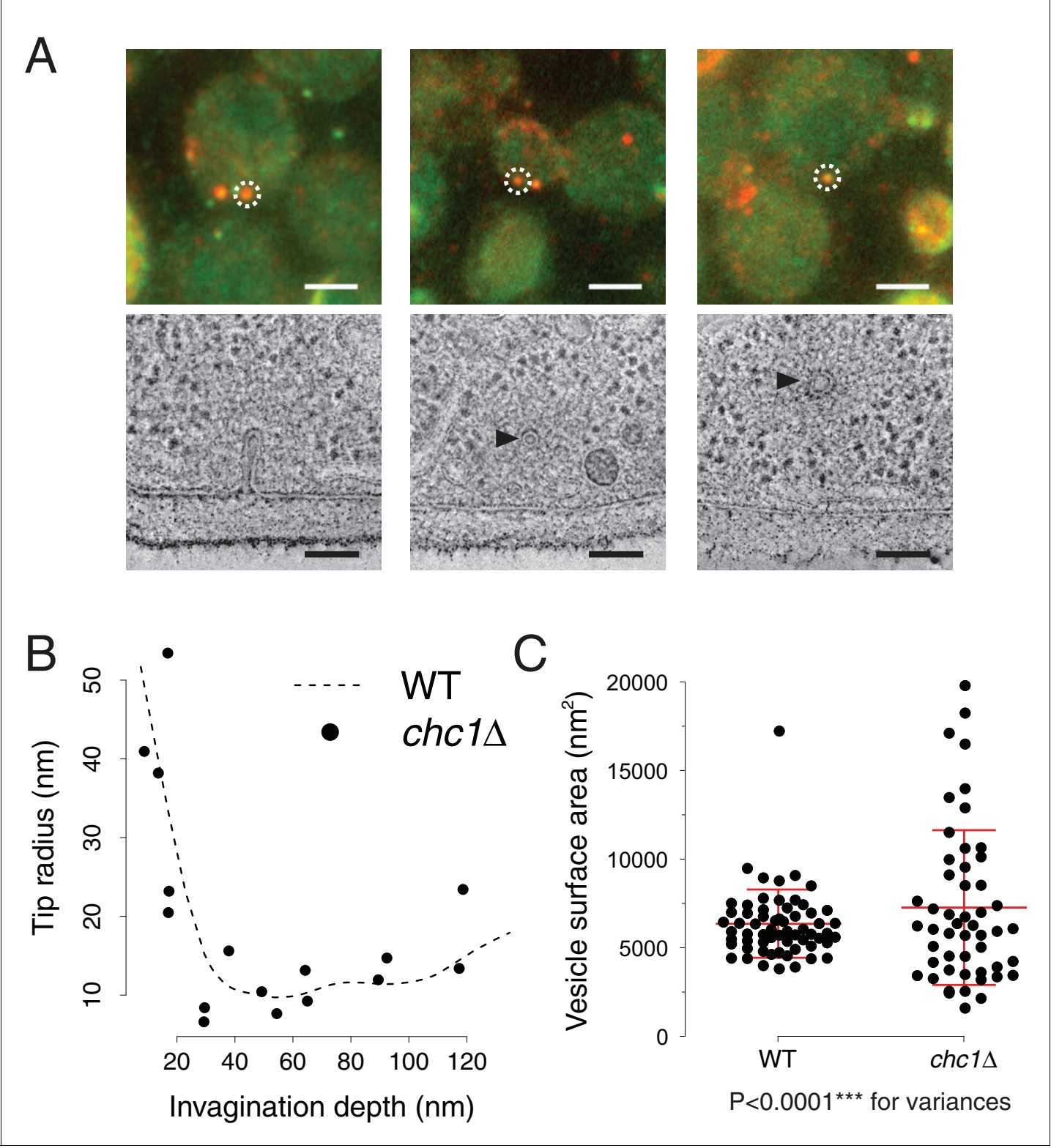

**Figure 2.** Endocytic membrane morphologies in *chc1Δ* cells. (**A**) Correlative fluorescence and electron microscopy of *chc1Δ* yeast cells expressing Sla1-GFP, Abp1-mCherry. Three examples of membrane ultrastructures underlying sites of Sla1 and Abp1 colocalization. Upper panel shows overlays of RFP and GFP channel images, lower panel images are virtual slices through electron tomograms, acquired at positions corresponding to the fluorescent spots marked by the white dashed circles. Black arrowheads indicate endocytic vesicles. Scale bars are 2 μm in the upper panel, 100 nm in the lower panel. (**B**) Curvature of invagination tips identified by correlative microscopy in *chc1Δ*, plotted against the invagination depths. The dashed line is a

*Figure 2 continued on next page*

*Figure 2 continued*

cubic smoothing spline fitted to the data points of wild-type invagination tip curvatures, published in (*Kukulski et al., 2012a*). (C) Surface areas of vesicles identified by correlative microscopy in *chc1Δ* and compared to wild-type cells (*Kukulski et al., 2012a*). Red central line represents the mean, upper and lower red lines represent standard deviations.

The following figure supplement is available for figure 2:

**Figure supplement 1.** Membrane shape parameters in *chc1Δ* cells.

MATa/MATα his3-Δ200/his3-Δ200 leu2-3,112/leu2-3,112 ura3-52/ura3-52 lys2-801/lys2-801 SLA1-EGFP::HIS3MX6/SLA1-EGFP::HIS3MX6  ABP1-mCherry::KanMX/ABP1-mCherry::KanMX  CHC1/chc1Δ::natNT2

### For live imaging

chc1Δ, Sla1-EGFP (MKY3020):
MATa/MATα his3-Δ200/his3-Δ200 leu2-3,112/leu2-3,112 ura3-52/ura3-52 lys2-801/lys2-801 SLA1-EGFP::HIS3MX6/SLA1 CHC1/chc1Δ::natNT2
chc1Δ, Rvs167-EGFP (MKY3022):
MATa/MATα  his3-Δ200/his3-Δ200  leu2-3,112/leu2-3,112  ura3-52/ura3-52  lys2-801/lys2- 801 RVS167-EGFP::HIS3XM6/RVS167 CHC1/chc1Δ::natNT2

## Correlative microscopy

Correlative fluorescence microscopy (FM) and electron tomography (ET) procedures were essentially as previously published (*Kukulski et al., 2011*, *2012b*). After sporulation of strain MKY2800, cultures of haploid *chc1Δ*, Sla1-GFP, Abp1-mCherry were grown for a maximum of 2 days before high-pressure freezing. Cultures grown at 25°C in SC-Trp medium to exponential phase were pelleted by vacuum filtration and high-pressure frozen using a Bal-tec HPM010 (*McDonald, 2007*). Freeze substitution, embedding in Lowicryl HM20 and sectioning was performed as before (*Kukulski et al., 2012a*). We used 50 nm TetraSpeck beads (Life Technologies), diluted 1:50 in PBS, as fiducial markers for correlation between FM and ET (*Suresh et al., 2015*). FM of grids was done on the same day, as described before (*Kukulski et al., 2012b*; *Suresh et al., 2015*). Grids incubated with 15 nm Protein A covered gold beads and stained with Reynold's lead citrate were subjected to ET exactly as described before (*Kukulski et al., 2012a*), using SerialEM for automatic tilt series acquisition and IMOD for reconstruction (*Mastronarde, 2005*; *Kremer et al., 1996*). The correlation procedure was performed using the in-house written procedure based on the Matlab Control Point Selection Tool (*Kukulski et al., 2011*). Positions of GFP and mCherry spots of interest were predicted by correlating TetraSpecks visible in average images of selected virtual slices of low-magnification tomograms to TetraSpeck signals in the respective channels. For further transformation into high-magnification tomograms, the 15 nm gold beads were used.

## Membrane ultrastructures in the correlative microscopy data set

We performed correlative microscopy on *chc1Δ* cells expressing Sla1-GFP and Abp1-mCherry. Like in wild-type cells, Sla1 in *chc1Δ* assembles at endocytic sites earlier than Abp1. The two proteins colocalize for several seconds, before Sla1 disassembles, followed by Abp1 a few seconds later (*Kaksonen et al., 2005*). We reconstructed 31 endocytic sites at which Sla1 and Abp1 colocalized. All these sites corresponded to ribosome exclusion zones that represent the endocytic machinery including the actin network (*Kukulski et al., 2012a*). Twelve contained invaginations, 15 contained vesicles, 3 contained invaginations and vesicles and 1 was a ribosome exclusion zone on a flat plasma membrane, containing no invagination or vesicle. Of the 28 sites at which Abp1 was present and Sla1 absent, 21 contained vesicles in ribosome exclusion zones and 7 were ribosome exclusion zones on a flat plasma membrane.

## Endocytic sites with multiple membrane ultrastructures in one actin network

In our correlative microscopy data set on *chc1Δ*, we also found an enrichment of sites consisting of more than one endocytic structure, as compared to the wild-type, such as two vesicles or two invaginations within one ribosome exclusion zone. In wild-type cells, about 5% of fluorescent spots consisted of multiple endocytic structures (*Kukulski et al., 2012a*). In *chc1Δ* cells, 13 of 59 Abp1-mCherry or Abp1-mCherrry/Sla1-GFP spots contained more than one endocytic membrane structure (22%). One of these sites contained 2 invaginations of different depths, 3 contained one invagination and one vesicle each, 6 contained 2 vesicles each and 3 contained 3 vesicles each. All these membrane ultrastructures were included in the quantitative analysis of membrane shapes, thus the presented data contains membrane structures from both single and multiple sites. For most of the parameters, there was no difference between single and multiple sites. However, the average surface area of vesicles from multiple endocytic sites is smaller than of vesicles from single endocytic sites in *chc1Δ* (*chc1Δ* 'multiple site vesicles': mean 5'720 $nm^2$, SD 2'688 $nm^2$, n = 24. *chc1Δ* 'singles': mean 8'688 $nm^2$, SD 5'106 $nm^2$, n = 27. p = 0.0118). They are similar to vesicles in wild-type cells (*chc1Δ* 'multiple site vesicles': mean 5'720 $nm^2$, SD 2'688 $nm^2$, n = 24. wt: mean 6'380 $nm^2$, SD 1'929 $nm^2$, n = 62. p = 0.2803 for mean). Furthermore, when only comparing *chc1Δ* vesicles from single endocytic sites to wild-type vesicles, there is a difference in their average surface areas (*chc1Δ* 'singles': mean 8'688 $nm^2$, SD 5'106 $nm^2$, n = 27. wt: mean 6'380 $nm^2$, SD 1'929 $nm^2$, n = 62. p = 0.0301 for mean). Two-tailed, unpaired t-tests with Welch's correction (assuming non-equal SD) were used to calculate the p values.

## Measurement of membrane parameters

To measure membrane parameters of invaginations and vesicles, we applied the same procedures as for the wild-type data set (*Kukulski et al., 2012a*). Thus, the parameters extracted from the *chc1Δ* dataset could be directly compared to the wild-type data. In brief, we used the EM package in Amira (*Pruggnaller et al., 2008*) to click points on the cytosolic leaflet of all invaginating membranes in a virtual tomographic slice showing the long axis of the invagination. The sets of points were aligned to an x-axis that represents the plasma membrane, and interpolated with a local second-degree polynomial fit in MATLAB. We used this data to determine the invagination depth, appearance of tubular segments, radius of the invagination tip and the position of the invagination neck (*Kukulski et al., 2012a*). The Amira EM package was also used to click points on the cytosolic leaflet of vesicle membranes within the tomographic volume. The points were used to fit an ellipsoid (MATLAB) and extract the three major axes from the vesicle. These were used to calculate the vesicle membrane surface area (*Kukulski et al., 2012a*). We used GraphPad Prism for statistical analysis and comparisons of data sets.

## Live fluorescence microscopy

Sporulated haploid strains were grown to log phase in SC-Trp medium and were adhered on a 25 mm coverslip that was first incubated for 10 min at room temperature with 40 µl of Concanavalin A (100 µg/ml) and then washed with SC-Trp medium. Cells were then imaged on the coverslip in 40 µl of SC-Trp at room temperature. The imaging was done with an Olympus IX81 inverted microscope equipped with an Olympus 100 x/1.45 NA TIRF objective, a GFP-3035C-OMF single-band filter set (Semrock) and a Hamamatsu ImagEM EMCCD set at full gain. Cells were excited for 80 ms (Rvs167-GFP) or 100 ms (Sla1-GFP) with a 488 nm laser.

## Image analysis

Images were processed with ImageJ. They were corrected for photobleaching and the local background was corrected by subtracting from the image the same image processed by a median filter with a kernel of 6 pixels (*Picco et al., 2015*). The spots were tracked using the Particle Tracker plugin in ImageJ (*Sbalzarini and Koumoutsakos, 2005*). In wild-type and *chc1Δ* cells, spots were tracked with the aim of not losing the dim phases at the beginning or at the end of the proteins lifetimes. The few fluorescent patches that showed abnormal brightness heights were discarded to reduce the chance of tracking sites with multiple endocytic events.

## Alignment of the trajectories

Sla1 trajectories from wild-type and *chc1Δ* cells were aligned in time and averaged as in (*Picco et al., 2015*). Rvs167 fluorescence intensities were scaled and aligned in time by searching for the minimum mean square displacement between the discrete integrals of the first 3 s of pairs of fluorescence intensity curves scaled by a factor $c$ and shifted in time by $\tau$: Let $\tilde{F} = \{\tilde{F}_j\}$ be a Rvs167 fluorescence intensity curve chosen as reference and $\tilde{f} = \{\tilde{f}_j\}$ one of the remaining fluorescence intensity curves that has to be aligned and scaled to $\tilde{F}$. $j$ indexes different time points. The discrete integrals of the fluorescence intensity values from the beginning of the trajectory until the time point $i$ are

$$F_i = \sum_{j=0}^{i} \tilde{F}_j \Delta t$$
$$f_i = \sum_{j=0}^{i} \tilde{f}_j \Delta t,$$

where $\Delta t$ is the amplitude of the time interval between consecutive time points. The different integrals were computed using only the fluorescence intensity values up to the first 3 s of the trajectories ($0 \le i \le 3$), which correspond roughly to the time it takes for Rvs167 to assemble on the invagination in wild-type cells. Each integral was then assigned to a time point $t$, which initially corresponded to the time indexed by $i : F_i = F_i[t]$ and $f_i = f_i[t]$, $i = t$. The time alignment $\tau$ and the scaling $c$ between the two fluorescence intensity curves were then defined as

$$c = \operatorname*{argmin}_c \left\langle (F_i[t] - c f_i[t+\tau])^2 \right\rangle = \frac{\sum_t F_i[t] f_i[t+\tau]}{\sum_t f_i[t+\tau] f_i[t+\tau]},$$

$$\tau = \operatorname*{argmin}_\tau \left\langle (F_i[t] - c f_i[t+\tau])^2 \right\rangle.$$

## Acknowledgements

WK acknowledges postdoctoral fellowships from the Swiss National Science Foundation and funding by the Medical Research Council (MC_UP_1201/8). Work in JAGB's lab was supported by the Chica und Heinz Schaller Stiftung. Work in MK's lab was supported by the Swiss National Science Foundation and the NCCR in Chemical Biology. This study was supported by the EMBL electron microscopy facility.

## Additional information

### Funding

| Funder | Grant reference number | Author |
|---|---|---|
| Schweizerischer Nationalfonds zur Förderung der Wissenschaftlichen Forschung | Postdoctoral Fellowships | Wanda Kukulski |
| Medical Research Council | MC_UP_1201/08 | Wanda Kukulski |
| Chica and Heinz Schaller Stiftung | | John AG Briggs |
| Schweizerischer Nationalfonds zur Förderung der Wissenschaftlichen Forschung | | Marko Kaksonen |
| National Centre of Competence in Research Chemical Biology | | Marko Kaksonen |

The funders had no role in study design, data collection and interpretation, or the decision to submit the work for publication.

## Author contributions
WK, AP, JAGB, MK, Read and approved the manuscript, Conception and design, Acquisition of data, Analysis and interpretation of data, Drafting or revising the article; TS, Read and approved the manuscript, Acquisition of data, Analysis and interpretation of data

## Author ORCIDs
Wanda Kukulski, http://orcid.org/0000-0002-2778-3936
Andrea Picco, http://orcid.org/0000-0003-2548-9183

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
