## [Decision Letter]

Thank you for submitting your article "Revisiting the role of clathrin in yeast endocytosis: Clathrin modulates vesicle scission, but not invagination shape" for consideration by *eLife*. Your article has been reviewed by three peer reviewers, and the evaluation has been overseen by Reviewing Editor Suzanne Pfeffer and Vivek Malhotra as the Senior Editor.

The reviewers have discussed the reviews with one another and the Reviewing Editor has drafted this decision to help you prepare a revised submission.

Summary:

This submission reports a correlative light and electron microscopy analysis of the dynamics of endocytic vesicle formation in *chc1*∆ yeast cells. The authors report that *chc1*∆ does not affect elongation or morphology of endocytic invaginations. However they describe significant persistence of Rvs167 at endocytic invaginations and a greater variability in the size of endocytic vesicles in *chc1*∆ cells. Based on these findings the authors propose that clathrin is not required for membrane invagination but instead regulates vesicle scission and size. Although there are some issues with interpretation of the data, the findings generally support the conclusions and will be of general interest. The work is appropriate as a Research Advance because it extends a previous study published in *eLife* (Picco et al., v4:e4535) by adding correlative light and EM analysis to investigate the question of clathrin function, which was not considered in the published work.

Essential revisions:

1) The authors conclude in the Results and Discussion section (first paragraph) that Sla1-GFP centroid movement in chc1∆ cells is indistinguishable from that in wild-type. However, in Figure 1, although the rate of Sla1 inward movement is the same, there is a clear extension of movement in *chc1*∆ cells. This difference should be noted and addressed in the text. The observation is consistent with data in Idrissi et al. (Figure 2, PNAS 109:2587-94. 2012) that shows an increase in Sla1 immuno-gold labeling of the longest invaginations in *chc1*∆ compared to wild-type cells. Considering the apparent defect in scission in *chc1*∆ cells, longer Sla1 invaginations are not particularly unexpected and do not present a challenge the authors' model.

2) The authors conclude from the data in Figure 2—figure supplement 1 that the detailed morphologies of membrane invaginations are the same in mutant and wild-type cells (Results and Discussion, fourth paragraph). In panel A, the equivalence of membrane angle versus invagination depth in mutant and wild-type cells is not entirely convincing. With the available data points, there could be a trend towards tubulation at shorter invagination stages in the mutant cells. Statistical analysis of the data or analysis of more mutant profiles could address this concern. An alteration in one aspect of invagination morphology does not detract from the striking anomaly in Rvs167 (scission) behavior in the absence of clathrin.

3) It is plausible that clathrin may still contribute to membrane sculpting in cells that are not subject to the same turgor pressure as yeast (and do not rely as heavily on actin for invagination). The authors should acknowledge this possibility.

4) There are many possible factors that can cause a change in vesicle scission and size and the authors do not show a direct link between clathrin and the role of Rvs167 in vesicle scission. Please state this explicitly. Additionally, the authors study endocytosis in a clathrin knockout cell line, but do not discuss how sick *chc1*∆ cells and how this may affect their results.

5) In the second paragraph of the Introduction, it is stated that there is diminished vesicle budding and receptor internalization in the *chc1*∆ background. Please provide a quantitative indication of the magnitude or extent of this decrease.

6) Please state explicitly whether the number of Sla1-GFP and Rvs167-GFP puncta that form over a given time interval in the *chc1*∆ yeast is equal to WT yeast. If not, the relative appearance frequency should be provided.

7) In Figure 2—figure supplement 1, the number of filled circles (*chc1*∆) versus open circles (WT) suggests that the endocytic structures in *chc1*∆ strains are more infrequent than in WT. Why?

8) The data presented and the conclusions drawn do not provide a coherent or explicit explanation for the slowed endocytosis in *chc1*∆ strains. The text does not discuss whether this is due to decreased nucleation or slowed maturation. The reader is left to puzzle this out for her/himself. In Figure 1, is the slowed loss of Rvs17-GFP in the *chc1*∆ stain sufficient to explain diminished endocytosis because of a limiting pool of Rvs167 since it is delayed at abnormal sites in *chc1*∆ cells?

9) The fourth paragraph of the Results and Discussion states that "… the rate at which the invagination (marked by Sla1-GFP) grows, as well as the detailed morphology of the membrane invaginations observed by electron tomography, were the same in wild type and *chc1*∆ cells." But this only pertains to the ones that they see and score. If the overall number of Sla1 puncta is decreased, does this perhaps argue for another earlier function for clathrin?

10) If the main point is that Rvs167 delays at endocytic patches cause somewhat heterogeneously sized CCVs, please explain or propose how the overlying clathrin coat interfaces with this amphiphysin orthologue to prompt its discharge. Do Maribel Geli's EM mapping studies put the assembled clathrin cap in close proximity with the Rvs167 ring? Please mention that vertebrate amphiphysin binds to both clathrin and AP-2 and so it might be worth looking for similar functional properties in Rvs167.

---

## [Author Response]

Essential revisions:

1) The authors conclude in the Results and Discussion section (first paragraph) that Sla1-GFP centroid movement in chc1∆ cells is indistinguishable from that in wild-type. However, in Figure 1, although the rate of Sla1 inward movement is the same, there is a clear extension of movement in chc1∆ cells. This difference should be noted and addressed in the text. The observation is consistent with data in Idrissi et al. (Figure 2, PNAS 109:2587-94. 2012) that shows an increase in Sla1 immuno-gold labeling of the longest invaginations in chc1∆ compared to wild-type cells. Considering the apparent defect in scission in chc1∆ cells, longer Sla1 invaginations are not particularly unexpected and do not present a challenge the authors' model.

Thank you for pointing this out. As requested we have now noted and addressed the observed difference in the text. As you have pointed out, while the rate of movement is the same, the extent of Sla1 centroid movement is larger in *chc1*∆ cells. This data is consistent with the increased immunolabelling seen in Figure 2 of the Idrissi et al., 2012 paper. However, neither we, nor Idrissi et al., see any difference in the distribution of invagination depths (see Introduction, first paragraph and Results and Discussion, third paragraph). This, together with the fact that the movement of Sla1 does not end with scission, but continues with the vesicle (Picco et al., 2015), makes it more likely that the greater extent of centroid movement is explained by slowed disassembly of Sla1-GFP (Figure 1—figure supplement 1) allowing us to track patches further into the cytoplasm.

“[…] and Sla1-GFP disassembly at the end of the endocytic process was subtly slowed down (Figure 1—figure supplement 1). The Sla1-GFP centroids moved slightly further into the cell than in wild-type cells (Figure 1), probably as a consequence of the slowed disassembly of Sla1-GFP that allowed us to track the patches in the cytoplasm for longer.”

*2) The authors conclude from the data in Figure 2—figure supplement 1 that the detailed morphologies of membrane invaginations are the same in mutant and wild-type cells (Results and Discussion, fourth paragraph). In panel A, the equivalence of membrane angle versus invagination depth in mutant and wild-type cells is not entirely convincing. With the available data points, there could be a trend towards tubulation at shorter invagination stages in the mutant cells. Statistical analysis of the data or analysis of more mutant profiles could address this concern. An alteration in one aspect of invagination morphology does not detract from the striking anomaly in Rvs167 (scission) behavior in the absence of clathrin.*

As requested, we have performed statistical analysis of the data shown in Figure 2—figure supplement 1, showing that the difference between *chc1*∆ and wild type is not significant (p value > 0.1), thus, there is no significant trend towards tubulation at shorter invaginations.

We further realized that due to superposition of wild type and *chc1*∆ data points, not all wild type data points were visible, making the similarity less convincing. We have now changed the style of data point symbols so that all data points are clearly visible.

3) It is plausible that clathrin may still contribute to membrane sculpting in cells that are not subject to the same turgor pressure as yeast (and do not rely as heavily on actin for invagination). The authors should acknowledge this possibility.

Indeed, that is possible. We have added the following sentence to comment on this point:

“Clathrin has been shown to induce membrane curvature in vitro (Dannhauser and Ungewickell, 2012), and we cannot rule out that it has a membrane sculpting role in other vesicle budding events, for instance when the membrane is subjected to less pressure than the turgor of the yeast cell.”

4) There are many possible factors that can cause a change in vesicle scission and size and the authors do not show a direct link between clathrin and the role of Rvs167 in vesicle scission. Please state this explicitly.

We have added the following sentence to make clear that our study does not show any direct link between Rvs167 and clathrin.

“How clathrin modulates disassembly of Rvs167 is an open question:”

This statement then leads into the discussion on how clathrin and Rvs167 may interact, as requested in point 10.

Additionally, the authors study endocytosis in a clathrin knockout cell line, but do not discuss how sick chc1∆ cells and how this may affect their results.

The *chc1*∆ cells are indeed sick: Besides a reduced rate of endocytic uptake, they exhibit a strong growth phenotype and an accumulation of internal membrane structures. This phenotype has been described in detail before (Payne et al., 1985, Payne et al., 1988, Chu et al., 1996). By using a temperature sensitive allele of *CHC1*, it has also been shown that the reduced rate of endocytic uptake is an immediate and direct cause of clathrin deficiency, whereas the slowed cell growth and changed intracellular morphology occur later, and are likely therefore a downstream effect (Tan et al., 1992). To discuss this point, we have added the following sentences to the Introduction:

“Although deletion of clathrin heavy chain also causes a strong growth phenotype and accumulation of internal membrane compartments (Payne et al., 1988; Payne and Schekman, 1985), the reduction of receptor-mediated endocytosis is independent of these effects (Tan et al., 1993). The abnormal morphology and slow growth of clathrin deficient cells is likely a downstream, cumulative effect of disrupted clathrin-mediated vesicle formation at the trans-Golgi network, endosomes and plasma membrane (Newpher et al., 2005; Brodsky et al., 2001).“

5) In the second paragraph of the Introduction, it is stated that there is diminished vesicle budding and receptor internalization in the chc1∆ background. Please provide a quantitative indication of the magnitude or extent of this decrease.

As requested, we have added the percentage of decrease in number of endocytic events, which has been published before (Kaksonen et al., 2005, Newpher et al., 2006).

“Previous research has shown that upon deletion of clathrin heavy chain, there is a 73% reduction in the frequency of endocytic vesicle budding events (Kaksonen, Toret and Drubin, 2005; Newpher and Lemmon, 2006), which can explain the reduced rate of receptor-mediated uptake (Chu, Pishvaee and Payne, 1996; Payne et al., 1988).”

6) Please state explicitly whether the number of Sla1-GFP and Rvs167-GFP puncta that form over a given time interval in the chc1∆ yeast is equal to WT yeast. If not, the relative appearance frequency should be provided.

The number of endocytic events that form over a given time is reduced by 73% in *chc1∆* (Kaksonen et al., 2005), which we now comment in the first paragraph of the Introduction. This was measured observing events marked by Sla1. We observe a comparable reduction for Rvs167 events in *chc1*∆. This is consistent with the observation that all Sla1-GFP punctae were found to successfully complete endocytosis in clathrin deficient cells (Newpher et al., 2006).

7) In Figure 2—figure supplement 1, the number of filled circles (chc1∆) versus open circles (WT) suggests that the endocytic structures in chc1∆ strains are more infrequent than in WT. Why?

Endocytic budding events in *chc1*∆ are indeed less frequent than in wild type cells, as described in responses to points 5 and 6, and thus overall there are less endocytic structures.

Furthermore, using the marker proteins Sla1 and Abp1, we found a lower fraction of endocytic structures to be invaginations and a higher fraction to be vesicles in *chc1*∆ compared to wild type cells (Kukulski et al., 2012, see also the Materials and methods section, subsection “Membrane ultrastructures in the correlative microscopy data set”). This difference is likely due to the slowed disassembly of the marker proteins in *chc1*Δ cells (Figure 1—figure supplement 1), resulting in a higher number of vesicles being detected.

8) The data presented and the conclusions drawn do not provide a coherent or explicit explanation for the slowed endocytosis in chc1∆ strains. The text does not discuss whether this is due to decreased nucleation or slowed maturation. The reader is left to puzzle this out for her/himself. In Figure 1, is the slowed loss of Rvs17-GFP in the chc1∆ stain sufficient to explain diminished endocytosis because of a limiting pool of Rvs167 since it is delayed at abnormal sites in chc1∆ cells?

As discussed in the previous three points, there is reduced endocytosis in *chc1∆* cells. Previous work suggests that this reflects a role for clathrin in the earlier stages of endocytosis, prior to the recruitment of Sla1 and the initiation of vesicle budding (Kaksonen et al., 2005 and Newpher et al., 2006). It seems unlikely that this effect is caused by a limiting pool of Rvs167. Our main conclusions relate to the later stages of endocytosis – here the subsequent membrane bending steps proceed largely normally despite the absence of clathrin. Furthermore, there seems to be a role for clathrin in regulating scission. As described in the previous three points, we have better introduced the earlier role of endocytosis, and have tried to provide a more coherent overall explanation at the end of the manuscript as follows:

“Previous work showed that clathrin has a role in initiating the vesicle budding process. Our study shows it has an additional role in modulating endocytic protein disassembly and the timing or position of the scission event and thereby the sizes of the resulting endocytic vesicles. We found no role for clathrin in sculpting the membrane during endocytic invagination in budding yeast.”

9) The fourth paragraph of the Results and Discussion states that "… the rate at which the invagination (marked by Sla1-GFP) grows, as well as the detailed morphology of the membrane invaginations observed by electron tomography, were the same in wild type and chc1∆ cells." But this only pertains to the ones that they see and score. If the overall number of Sla1 puncta is decreased, does this perhaps argue for another earlier function for clathrin?

Yes, there is an earlier function for clathrin. The role of clathrin in the early steps of endocytosis is now better discussed in the manuscript, as described also in the responses to points 5 and 8 (Introduction, first paragraph and Results and Discussion, last paragraph):

“Taken together, previous studies suggest that clathrin plays a role in initiating vesicle budding.”

Additionally, we added a sentence at the beginning of the Results and Discussion section to make clear that the endocytic events we analyze are those that successfully initiate budding.

“Although fewer endocytic sites are observed in chc1∆ cells, Sla1 marks those that successfully initiate budding (Kaksonen, Toret and Drubin, 2005;Newpher and Lemmon, 2006).”

*10) If the main point is that Rvs167 delays at endocytic patches cause somewhat heterogeneously sized CCVs, please explain or propose how the overlying clathrin coat interfaces with this amphiphysin orthologue to prompt its discharge. Do Maribel Geli's EM mapping studies put the assembled clathrin cap in close proximity with the Rvs167 ring? Please mention that vertebrate amphiphysin binds to both clathrin and AP-2 and so it might be worth looking for similar functional properties in Rvs167.*

We have added a paragraph discussing the possibility of protein-protein interactions between amphiphysins and clathrin, referring to the mentioned immuno-EM study as well as studies reporting binding between clathrin and amphiphysins.

“How clathrin modulates disassembly of Rvs167 is an open question: The two proteins occupy adjacent regions on the endocytic invagination (Idrissi et al., 2008), and in vertebrates, amphiphysins bind directly to clathrin and the endocytic adaptor AP-2 (McMahon, Wigge and Smith, 1997; Slepnev et al., 2000). Thus, we speculate that protein-protein interactions involving clathrin could modulate the disassembly dynamics of Rvs167, either through direct binding or recruitment of other regulatory proteins.”